# Identification of Potential Antimalarial Drug Candidates Targeting Falcipain-2 Protein of Malaria Parasite—A Computational Strategy

**DOI:** 10.3390/biotech11040054

**Published:** 2022-11-30

**Authors:** Shrikant Nema, Kanika Verma, Ashutosh Mani, Neha Shree Maurya, Archana Tiwari, Praveen Kumar Bharti

**Affiliations:** 1Division of Vector-Borne Diseases, ICMR-National Institute of Research in Tribal Health, Jabalpur 482 003, Madhya Pradesh, India; 2School of Biotechnology, Rajiv Gandhi Proudyogiki Vishwavidyalaya (State Technological University of Madhya Pradesh), Bhopal 462 023, Madhya Pradesh, India; 3Department of Biotechnology, Motilal Nehru National Institute of Technology, Allahabad 211 004, Uttar Pradesh, India

**Keywords:** molecular docking, FP-2, alkaloids, antimalarial drug resistance, molecular dynamics simulation

## Abstract

Falcipain-2 (FP-2) is one of the main haemoglobinase of *P. falciparum* which is an important molecular target for the treatment of malaria. In this study, we have screened alkaloids to identify potential inhibitors against FP-2 since alkaloids possess great potential as anti-malarial agents. A total of 340 alkaloids were considered for the study using a series of computational pipelines. Initially, pharmacokinetics and toxicity risk assessment parameters were applied to screen compounds. Subsequently, molecular docking algorithms were utilised to understand the binding efficiency of alkaloids against FP-2. Further, oral toxicity prediction was done using the pkCSM tool, and 3D pharmacophore features were analysed using the PharmaGist server. Finally, MD simulation was performed for Artemisinin and the top 3 drug candidates (Noscapine, Reticuline, Aclidinium) based on docking scores to understand the functional impact of the complexes, followed by a binding site interaction residues study. Overall analysis suggests that Noscapine conceded good pharmacokinetics and oral bioavailability properties. Also, it showed better binding efficiency with FP-2 when compared to Artemisinin. Interestingly, structure alignment analysis with artemisinin revealed that Noscapine, Reticuline, and Aclidinium might possess similar biological action. Molecular dynamics and free energy calculations revealed that Noscapine could be a potent antimalarial agent targeting FP-2 that can be used for the treatment of malaria and need to be studied experimentally in the future.

## 1. Introduction

After significant efforts, malaria remains a major public health problem worldwide. More than 40% of the world’s population and nearly 90% of India’s population is at risk of malarial infection [1]. The failure of the antimalarial drug led to the re-emergence of malaria and the spread of the resistant parasite in malaria-endemic countries. The development of drug resistance to currently available antimalarial drugs has increased severity, eventually posing a major challenge to the malaria elimination goal [2]. The clinical symptoms of malaria are associated with blood-stage parasites and linked to a large-scale infection of erythrocytes [3]. Intraerythrocytic malaria parasites break down haemoglobin (around 60–80%) in an acidic food vacuole to supply amino acids for parasite protein synthesis and provide the space for the parasite to grow and replicate in the erythrocyte eventually creating a hindrance in parasite survivability by toxic heme [4]. As a survival tactic, the parasite uses a ~200 KDa protein complex containing cysteine falcipain-2 (FP-2), aspartate (plasmepsin II and IV), histo-aspartic proteases, and a dedicated enzyme (heme detoxification protein) for converting toxic heme to ‘hemozoin’ [5]. This protects the parasite from oxidative damage by reactive oxygen species (ROS). In this complex process, FP-2 is a papain-family member that belongs to the cysteine protease of the *P. falciparum* erythrocytic stage, localised in the food vacuole and hydrolyse the haemoglobin molecule. The gene on chromosome 11 encodes FP-2. FP-2 is expressed throughout the erythrocytic stages of the parasite lifecycle. However, comparatively higher expression was found in the trophozoite stage [6]. In the parasite intra-erythrocytic cycle, the skeletal protein ankyrin is degraded by FP-2 in the process of rupturing red blood cells [7]. FP-2 consists of two domains, the pro-domain and the mature domain. The pro-domain has an N-terminus consisting of the first 35 cytosolic amino acids, followed by a 20 amino acid transmembrane domain and a 188 amino acid lumenal domain. The C-terminal part of the Pro-domain consist of an inhibitory domain that contains two motifs, ERFNIN and GNFD, that bind to the active site and thus regulate the function of mature FP enzymes [8]. The mature domain of FP-2 is a 27 KDa protease that contains refolding domain, an active site cleft and is responsible for haemoglobin cleavage into small peptides [9]. Sijwali et al. reported that FP-2 has a crucial role in haemoglobin hydrolysis. However, the cysteine protease activity of Falcipain-2-knockout trophozoites was significantly reduced, indicating a block in haemoglobin breakdown [10]. Therefore, inhibition of the FP-2 enzyme could be a promising target for antimalarial drugs as it blocks haemoglobin hydrolysis and parasite development [8].

In the present scenario, resistance to currently available antimalarial drugs has been confirmed resistance in the two most predominant malaria parasite species, i.e., *P. falciparum* and *P. vivax*, which pose the greatest threat to malaria control and its elimination [11]. Liu et al. have shown the inhibitory action of artemisinin against FP-2 [12]. However, it has been reported that the FP-2 conferred a degree of Artemisinin resistance to the early rings stage [13,14]. This warrants an urgent need for designing and developing novel lead candidates, particularly from natural sources. Of note, alkaloids have been recognised as important phytoconstituents with interesting biological properties (anti-malarial, anti-cancerous, and anti-helminth) for many years. It is worth mentioning that the first successful antimalarial drug was quinine, an alkaloid, which was extracted from the Cinchona tree. Therefore, we have employed the virtual screening (VS) approach to distinguish the active molecules from the inactive ones [15]. VS approach refers to the computational screening of chemical libraries for compounds that are potential enough to target the biomolecules efficiently and effectively [16]. Thus, the binding affinity of alkaloids from the Drug Bank and NPACT database was checked against the FP-2 protein [17,18]. Finally, we performed a molecular dynamics simulation of the screened alkaloids that have shown maximum binding affinity against the FP-2 protein to understand the stability of the FP-2 structure upon binding of alkaloids.

## 2. Materials and Methods

To analyse the binding site on the surface of the FP-2, blind molecular docking was performed [19,20]. In this study, we performed the docking analysis of alkaloids from Drug Bank and Naturally Occurring Plant-based Anti-cancer Compound-Activity-Target database (NPACT) [21] along with reference drug molecule Artemisinin against FP-2. The flowchart of the present study is given in Figure 1.

### 2.1. Dataset

The 3-dimensional (3D) structure of FP-2 of *P. falciparum* was downloaded from Protein Data Bank (http://www.pdb.org, accessed on 2 September 2022). The PDB ID for FP-2 is 6JW9 (Resolution 3.50 A; Figure 2). The protein structure was then prepared by removing all water molecules and hetatoms. The compounds were taken from the NPACT database and the Drug Bank database [22]. The 3D structure of alkaloids and reference molecule Artemisinin (ID 68827) were retrieved from the PubChem database. One compound (Noscapine) was excluded due to duplicity in the drug bank and NPACT databases. Three compounds (Cryptenamine, IT-101, XMT-1001) were excluded due to the non-availability of the crucial details of the compounds. A total of 340 compounds were taken for analysis.

### 2.2. Absorption, Distribution, Metabolism, and Excretion (ADME) Analysis

ADME analysis was performed to carry out the pharmacokinetic analysis of alkaloids along with a reference molecule (Artemisinin) via the Molinspiration program (http://www.molinspiration.com, accessed on 27 November 2022). Artemisinin values were set as the threshold limit for screening the list of compounds from the database. The pharmacokinetic properties of alkaloids were calculated according to Lipinski’s Rule of Five (Ro5), the key molecular descriptors considered within the subsequent range; partition coefficient (logP) ≤ 5, molecular weight (MW) ≤ 500, hydrogen bond acceptors and donors ≤ 10 and ≤5 respectively, and Van der Waals bumps’ polar surface area (PSA) <120 Ǻ [23]. All these properties together were explored for all the molecules to evaluate their drug-like nature and whether they are likely to be orally bioavailable [24].

### 2.3. Toxicity Risk Analysis and Drug-Likeness Prediction

The data warrior program was used to perform the toxicity risk assessment and to predict the drug-likeness score. Chemical structures were used to predict physicochemical properties. Toxicity risk was predicted from the default set of parameters defined in an algorithm that gives rise to toxicity alerts [25]. It is the quantification of lethality, which involves four types of toxic effects such as mutagenicity, irritant, tumorigenic and reproductive effects of the alkaloids used in the dataset. Furthermore, drug-likeness is a qualitative aspect that is defined as the balance of various molecular properties and structural characteristics, which helps in determining the similarity with the known drugs for the drug design. It is determined by various properties such as hydrophobicity, hydrogen bonding characteristics, electronic distribution, molecule size and flexibility, presence of various pharmacophore characteristics which influence the nature of molecule in a living organism, including bioavailability, affinity to proteins, reactivity, toxicity, transport properties, and metabolic stability [26].

### 2.4. Protein-Ligand Docking Study

To determine the binding free energy of alkaloids towards FP-2 protein, molecular docking was accomplished via 2 different docking algorithms. Initially, iGEMDOCK was employed for the docking, followed by Auto dock vina (Chimera plug-in) analysis. iGEMDOCK is a fully integrated VS environment that includes everything from preparation to post-screening analysis, including pharmaceutical interactions. To begin, iGEMDOCK includes interactive interfaces for preparing both the target protein’s binding site and the screening compound library. Whereas Auto Dock Vina (http://vina.scripps.edu, accessed on 27 November 2022) (Chimera Plugged-in) is a program which works on the basis of a scoring function that can be seen as an attempt to approximate the standard chemical potentials of the system [27]. To be noted, the artemisinin docking scores were set as cut-off values for screening the drug molecules.

### 2.5. Oral Toxicity Assessment

The oral toxicity assessment for the alkaloids was performed by using the pkCSM tool [28]. The purpose of acute toxicity testing is to obtain information on the biological activity of a chemical and gain insight into its mechanism of action [29]. In the present study, prediction of Ames toxicity was done to assess the mutagenicity of compounds; oral rat acute toxicity (LD50) is the measurement of acute toxicity that causes the death of half of the group of test animals [30,31] oral rat chronic toxicity (LOAEL) identifies the exposure of low-moderate doses of chemicals over a long period of time [32] and hepatotoxicity of compounds were analysed. Compounds were considered hepatotoxic if they influenced the normal physiological function of the liver. It is worth mentioning that the oral toxicity of alkaloids evaluated by this platform is comparable to the toxicity risk study in the animal model study [28].

### 2.6. 3D Pharmacophore Analysis

Ligand-based 3D pharmacophore of drug molecules was derived using the PharmaGist web server. Based on the different conformations of the molecules, various pharmacophores were built by the web server in decreasing order of score. PharmaGist offers a quick, reliable, and quantitative technique to compare pharmacophore features between two ligands by utilising a deterministic and efficient algorithm for flexible pairwise and multiple alignments of the 3-D conformational structures of ligands [33].

### 2.7. Molecular Dynamics (MD) Simulation

Docked complex structures of FP-2-Artemisinin and FP-2-Noscapine, FP-2-reticuline, and FP-2-Aclidinium were used as the initial structures for the execution of the Molecular Simulation study. Gromacs version 2020.1 [34] was used for performing the molecular dynamics simulation (MDS) for FP-2 receptor (protein) with Artemisinin and Noscapine, Reticuline, Aclidinium (ligand) using GROMOS-54A7 force field [34]. Topology files were generated for the protein-ligand complex using the pdb2gmx module from the GROMACS package. TIP3P water model was used for the solvation of the complex system. Energy minimisation (EM) for each protein-ligand complex was performed using the steepest descent integrator with a maximum number of 50,000 steps. After the EM step, the protein-ligand complex was equilibrated under NVT (canonical ensemble) and NPT (isothermal-isobaric ensemble) conditions for 1 ns (1000 ps) at 300 K [35]. The production simulation run of 100 ns (100,000 ps) was performed for the protein-ligand complex system. Berendsen’s weak-coupling method was used for the maintenance of the temperature and pressure of the protein-ligand complex system [34]. Van der Waals interactions were calculated using the Lennard–Jones potential, and a particle-mesh Ewald was used for long-range electrostatic interactions with a cut-off of 1.2 nm. All the bonds were constrained using the LINCS algorithm [36]. Gromacs in-built tools were used for the MDS analysis, which includes the root-mean-square deviation (RMSD), root means square fluctuation (RMSF), solvent-accessible surface area (SASA), and radius of gyration (Rg). Xm grace tool was used for analysing the trajectories and graph plotting [37].

### 2.8. Binding Free Energy Calculations Using MM-PBSA Approach

The binding free energy of each FP-2-ligand complex was determined by employing the molecular mechanics Poisson–Boltzmann surface area (MM-PBSA) using MMPBSA package 41. MMPBSA is the most widely used approach to calculate the interaction energies among the protein-ligand complexes [38]. Together with MD simulation, MM-PBSA can decode significant conformational fluctuations and entropic contributions to the binding energy [39]. The trajectories were extracted from the last 20 ns time period, and the set of equations below was used to determine binding free energy (ΔG*_bind_*).
ΔG*_bind_* = ΔG*_complex_* − (ΔG*_receptor_* + ΔG*_ligand_*)(1)
where ΔG*_complex_* is the total free energy of the protein-ligand complex, and ΔG*_receptor_* and ΔG*_ligand_* are the total free energies of the separated protein and ligand insolvent, respectively.

## 3. Results

### 3.1. Pharmacokinetics Analysis

ADME analysis was performed using the Molinspiration program to calculate the pharmacokinetic properties of 340 alkaloids. The value that corresponds to Artemisinin was set as the threshold value for screening the alkaloids. The properties of the compounds are depicted in Appendix A. It is observed from the results that Artemisinin showed violations zero according to the Ro5. For instance, 76 compounds from all data sets were violating the Ro5 with 1 or >1 violations, whereas 264 compounds showed no violations according to the Ro5, signifying the enhanced pharmacokinetic properties of alkaloids. Therefore, a total of 264 compounds were further processed for the oral toxicity and drug-likeness assessment.

### 3.2. Toxicity Risk and Drug-Likeness

Drug likeliness and toxicity risk assessment test was done for 264 alkaloids along with artemisinin. The results are presented in Appendix A. Alkaloids scoring positive drug-likeness values were further subjected to toxicity risk assessment analysis. Altogether, 120 alkaloids indicating a range of positive drug-likeness values with no mutagenicity, no tumorigenicity, no reproductive effect, and no irritant properties were further considered for docking studies.

### 3.3. Biomolecules Docking Analysis

Alkaloids with good pharmacokinetic properties and drug-likeness were considered for molecular docking study. A total of 120 Alkaloids were docked against FP-2 protein using the iGEMDOCK docking algorithm to investigate the inhibitory action towards FP-2 protein. From the results, we found that 62 alkaloids have shown greater binding energy scores when compared to Artemisinin. Threshold docking scores of reference molecule (Artemisinin) for each docking program (iGEMDOCK and Autodock Vina) were set, i.e., −91.37 and −6.6 kcal/mol, to screen out the alkaloids. The results attained from the docking were tabulated in Appendix A. Thereafter, 62 alkaloids were further docked against FP-2 protein using auto dock vina (Chimera Plug-in). The results from the analysis indicated that 38 alkaloids had shown a good binding efficiency than Artemisinin. Both docking tools work on different algorithms. The results of auto dock vina (Chimera Plug-in) are represented in Figure 3. Taken collectively, both algorithms revealed that a total of 38 compounds had shown a higher binding affinity towards FP-2 as compared to artemisinin.

### 3.4. Oral Toxicity Assessment

The oral toxicity risks assessment of the 38 alkaloids with efficient binding energies along with Artemisinin was investigated using the Graph-Based Signatures program implemented in the pkCSM tool. The results revealed that a total of 11 compounds had no Ames toxicity and no hepatoxicity. However, artemisinin showed Ames toxicity. The results are tabulated in Appendix A. In addition, the oral rat acute toxicity (LD50) value and oral rat chronic toxicity (LOAEL) was also calculated for all the alkaloids (Appendix A) to gain insight into drug mechanism action and to understand the biological activity of a chemical [29]. Thus, all 11 compounds with no Ames toxicity and hepatoxicity were considered for the 3D pharmacophore study.

### 3.5. 3D Pharmacophore Study

PharmaGist was used for detecting shared 3D pharmacophore features by Artemisinin and all the alkaloids’ subsets. The program would help to predict the spatial arrangements of various chemical features like hydrogen bond acceptors (HBA), hydrogen bond donors (HBD), hydrophobic centres (HY) aromatic rings, hydrophobic areas, and positive or negative ionisable groups for the input alkaloids and artemisinin-based on multiple flexible alignments. These pharmacophore features of a molecule are important for a ligand interaction with the target protein. Results from the analysis revealed that a total of 11 compounds attained the Best Pairwise alignment score value based on structure alignment with artemisinin. However, the top six compounds, Noscapine, Reticuline, Aclidinium, Benzoylecgonine, Vincamine, and Tretoquinol, attained the highest pairwise alignment score value of 6.64027, 6.6333, 6.3347, 6.32994, 6.329, 6.32673 respectively. The results are tabulated in Appendix A. Thus, the study revealed that pharmacophore features shared between alkaloids and artemisinin were majorly among these identified six alkaloids. Among these six alkaloids, the top three (Noscapine, Reticuline, Aclidinium) with the highest pairwise alignment score were considered for MSD, and this can be further utilised for generating new potent lead candidates in the process of drug designing for the treatment of malaria.

### 3.6. Molecular Dynamics (MD) Simulation Analysis

MD simulation was accomplished for FP-2-Artemisinin, FP-2 Noscapine, FP-2 Reticuline, and FP-2 Aclidinium complex structures. From the trajectory files of the simulation study, the data for RMSD, RMSF, SASA, the radius of gyration, hydrogen bonds (H-Bonds), and binding free energy calculations using the MMPBSA method were studied. RMSD calculation was done for the entire C-α atom from the starting structures, which was considered an essential criterion to calculate the convergence of the protein-ligand complex system involved in the study.

Figure 4 suggests that the RMSD values of all the complex structures fluctuate between 0.2–0.5 nm; also, the RMSD values attained by FP-2-Artemisinin, FP-2 Noscapine, FP-2 Reticuline, and FP-2 Aclidinium systems is less than 2 nm. This indicates that all the systems-maintained equilibrium throughout the simulation. However, FP-2-Artemisinin and FP-2-Noscapine, the RMSD averaged between ~0.2–0.29 nm and ~0.32–0.39 nm, respectively, during the first 40 ns of the simulation. It then rose to ~0.31 for Artemisinin at a point around 42 ns, and later it was fairly stable until the 100 ns simulation run with the RMSD of ~0.25–0.29 nm. The RMSD attained by FP-2-Noscapine after 45 ns was quite stable as compared to Artemisinin and the other two candidates and showed a reasonable RMSD value of ~0.35 nm during the rest of the simulation period. The RMSD values attained by FP-2 Reticuline and FP-2 Aclidinium systems at the start of the simulation were in the range of ~0.25–0.32 nm. Hereafter, both the systems showed fluctuations till 20 ns and were observed in a similar trend until 55 ns and maintained an average RMSD of ~0.40 nm. FP-2 Reticuline RMSD then rose to ~0.45 nm after 60 ns, while FP-2-Aclidinium around 80 ns with RMSD value rose to ~0.41 nm. However, both systems were fairly stable in the last 20 ns of the simulation period. Further, to understand the dynamic behaviour of the residues, we have demonstrated the RMSF of the amino acid residues of the complex structures.

Analysis of residue fluctuation depicts that the RMSF for all the complex structures was in a similar pattern. Off note, the higher fluctuating amino acid residues were from position 175 to 200. Overall, the graph suggests that FP-2-Reticuline complex residues showed higher fluctuations throughout the simulation period when compared to FP-2-Artemisinin, FP-2 Noscapine, and FP-2 Aclidinium complexes (Figure 5). Moreover, the residues of FP-2-Reticuline attained the highest flexibility over other complex structures. To be noted, the residues 175–200 showed the highest fluctuation with increasing order of RMSF values as FP-2 Noscapine > FP-2 Aclidinium > FP-2-Reticuline > FP-2-Artemisinin.

Subsequently, we performed the geometry and surface analysis for all four complex structures by plotting the graphs for SASA (Figure 6) and the radius of gyration (Rg; Figure 7). All four systems (FP-2-Artemisinin, FP-2 Noscapine, FP-2 Reticuline, and FP-2 Aclidinium) showed similar trends for the first 30 ns of the simulation run with reasonable SASA values in the range of 120–135 Å. Hereafter, a slight deviation was observed at 40 ns for the FP-2 Reticuline system with an RMSD value of ~135 Å. Later, FP-2 Reticuline and FP-2 Noscapine showed a similar pattern of the SASA from 45 ns to 64 ns of simulation with the SASA values attained in the range of ~120–128 Å. Further, FP-2 Noscapine showed a slight deviation around 65 ns of the simulation and attained the SASA values of ~125 nm until the end of the simulation, whereas FP-2-Artemisinin, FP-2 Reticuline, and FP-2 Aclidinium systems showed a similar pattern of SASA after 65 ns till the end of simulation with the SASA values in the range of ~125–132 Å (Figure 6). Lesser SASA values for the FP-2-Noscapine complex indicate that the protein is not unfolded and is less exposed to the solvent. However, the receptor of FP-2-Artemisinin, FP-2 Reticuline, and FP-2 Aclidinium complexes was unfolded and exposed the underlying hydrophobic residues to the solvent. It is noteworthy that the lesser accessibility of the FP-2-Noscapine complex by a number of water molecules will result in good binding efficiency when compared to the other three complex systems, including the reference drug Artemisinin. The results for Rg calculations correlate well with SASA analysis. At the start of the simulation, the Rg values attained by the FP-2-Artemisinin, FP-2 Aclidinium, and FP-2 Noscapine and FP-2 Reticuline complex structures lie in the range of 1.8 nm to 1.9 nm which is quite similar till 20 ns. The Rg values were higher for the FP-2 Aclidinium system when compared to the other three complexes with Rg values of ~1.91 nm to 80 ns. Afterwards, the Rg values decreased to ~1.85 nm till the end of the simulation. The Rg values for the FP-2-reticuline complex are quite stable for the period of 20 ns–50 ns, with the Rg values attained at ~1.87 nm. Later, slight fluctuations were observed till the end of the simulation. Similarly, the FP-2-Artemisinin system showed fluctuations at different points of the simulation run after 25 ns and continued till 65 ns. But later, after 70 ns, the Rg was quite stable till 100 ns with reasonable Rg values of ~1.85 nm. However, Rg for FP-2 Noscapine observed was ~1.8 nm which is quite less than the other three systems. In addition, more fluctuations were noticed for the Noscapine system during the time period of 20 ns–50 ns, with the Rg value in the range of 1.8 nm–1.85 nm. After 55 ns, the system was equilibrated and showed stability until 100 ns of the simulation run with Rg values attained ~1.77 nm (Figure 7). Overall, the Rg results suggest that Artemisinin, Aclidinium, and Reticuline binding to FP-2 decrease the stability of the protein structure when compared to the Noscapine. Furthermore, to understand the contribution and involvement of micro factors in maintaining the binding affinity between the protein and a drug molecule complex, Hydrogen Bonds (H-bonds) were analysed between molecules for all the complex structures of FP-2 in the simulation period of 100 ns. FP-2-Artemisinin was bound to the FP-2 pocket with two H-bonds during the simulation period of 15–25 ns. Later, it maintained only one H-bond with the active site residues, while FP-2-Noscapine, FP-2-Reticuline and FP-2-Aclidinium were able to form an average of 2–3 H-bonds throughout the simulation period. (Figure 8). This shows that H-bond is contributing to the strong binding of Noscapine, Reticuline and Aclidinium towards FP-2 protein. These results also indicate that the top three candidates have a great calibre to function as a strong inhibitor of the FP-2 protein.

### 3.7. Binding Free Energy Calculations of FP-2 Complex Proteins

For ΔG*_bind_* calculations were processed by extracting the MD trajectories from the last 20 ns of the simulation for all the four systems, i.e., FP-2-Artemisinin, FP-2 Noscapine, FP-2 Reticuline, and FP-2 Aclidinium. The results from the analysis reveal that Vander Waal energy was the main stabilising energy for the binding Artemisinin, Noscapine, Reticuline and Aclidinium (Appendix A). The estimated ΔG*_bind_* for FP-2-Artemisinin, FP-2 Noscapine, FP-2 Reticuline, and FP-2 Aclidinium based on the MMPBSA method was −64.76 +/− 10.39, −104.44 +/− 13.80, −183.37 +/− 10.88 and −204.23 +/− 12.37 kJ/mol, respectively. Overall, the results indicated that FP-2 Noscapine, FP-2 Reticuline, and FP-2 Aclidinium showed lower ΔG*_bind_* than the FP-2-Artemisinin system, suggesting that they have a higher binding affinity towards FP-2 Protein.

### 3.8. Alkaloids Interaction Study with FP-2

Interaction studies were carried out for the top three alkaloids along with Artemisinin using Biovia Discovery Studio (Biovia Discovery Studio). From the interaction study, we observed that Artemisinin could form hydrophobic interactions with TRP206 amino acid residue of FP-2, whereas Reticuline is also able to form hydrophobic interactions with TRP206 and hydrogen bonding with ASN173 residue. Noscapine is able to form pi-pi stacking with TRP206 along with Vander Waal bond and hydrogen bond, and pi-alkyl interaction with ALA157. Aclidinium formed a conventional Hydrogen bond with GLN36 (Figure 9). Moreover, other residues participate in the efficient binding of alkaloids, such as ASP234, SER149, LEU84, HIS174, GLN36, ASP35, ASN173, ALA157, TRP206, LYS37, GLN209, ASN38, PHE156, and VAL152 (Appendix A). Notably, catalytic sites of falcipain consist of four different pockets S1, S2, S3, and S1′. Among these, S2 is the most well-defined pocket and favours the binding of substrate, which comprises residues ASP234, SER149, PHE236, and ILE85 in the FP-2. This suggests that alkaloids bind near the S2 pocket and is quite similar to the binding pattern of E64 [40].

## 4. Discussion

FP-2 plays an important role in the parasite life cycle. Inhibition of FP-2 prevents parasite maturation and could be a valuable target for designing novel antimalarial drugs [9]. A study by Rosmalena et al. has shown that cinchona alkaloid derivatives have shown inhibitory action against FP-2 [41]. Uddin et al. have also shown the inhibitory action of falcipain-2 against natural compounds [42]. In the present study, we have used 340 plant-derived alkaloids against FP-2 to identify the potential antimalarial agents. The analysis from the current study revealed that 264 compounds possess good pharmacokinetic properties as compared to Artemisinin. To be a drug-like candidate, a molecule satisfies pharmacokinetic parameters, which is the prime reason for the failure of 50% of drugs in clinical trials [43], so it is important to eliminate the compounds in the early stage of drug discovery. Also, the pharmacokinetic profile of a compound greatly influences the effectiveness of drugs in the body [44]. Further, these 264 compounds were subjected to be screened for toxicity risk and drug-likeness assessment. According to the results, there were 120 compounds with good bioavailability and no toxicity features. The positive values of drug-likeness indicate that the compound contains mainly the fragments which are usually present in the available market drugs [45]. Later, these compounds were considered for docking studies to understand the binding efficiency. The results of the docking analysis suggest that 38 alkaloids have scored high binding energies, which indicates good binding affinity towards FP-2. For instance, the highest binding energy of receptor-ligand interaction supports the fitting of the drug to the target molecules. The larger the negative value of binding energy, the greater the chemical be accepted as a drug. The higher negative values imply the stability of the complex, and greater will be the propensity of the alkaloids to associate with FP-2 [46]. Furthermore, oral toxicity and oral absorption evaluation results indicate that 11 alkaloids were in the comparable zone. To be noted, toxicological screening is an important aspect to initiate the clinical investigation of new drugs [47]. Based on structural alignment, 3D pharmacophore analysis revealed that Noscapine, Reticuline, Aclidinium, Benzoylecgonine, Vincamine, and Tretoquinol shared pharmacophore features with Artemisinin. Understanding the mechanics and energetics of ligand binding is an inaccessible task using experimental techniques, as the binding of the ligand is a microscopic event that could be addressed via a molecular simulation approach [48]. Molecular simulation studies revealed that the Noscapine system was quite stable and equilibrated well after 50 ns of the simulation run and exhibited insignificant conformational changes in protein structure when compared to Reticuline, Aclidinium, and reference drug Artemisinin. Also, these three systems attained stability towards the end of the simulation around 80 ns. Further, to understand the dynamic behaviour of the residues we have demonstrated the RMSF of the amino acid residues of the complex structures. It is observed from the RMSF results that the amino acid residue of Artemisinin, Aclidinium and Noscapine showed lesser fluctuation and are less flexible when compared to Reticuline. Thus, it indicates that the Artemisinin, Aclidinium and Noscapine are strongly bound inside the active site with the residues of FP-2 protein. The higher flexibility of the residues from FP-2-Reticuline system is evidence of the lesser involvement of those residues in the binding towards the protein [49]. Moreover, SASA analysis was performed for all the top 4 complexes along with Artemisinin, as the drug interaction with the target protein could be affected by the water molecules’ accessibility [50]. From the SASA results, we observed that the FP-2-Noscapine complex was less accessible to the water molecules when compared to other systems, which states that the Noscapine binding is not affected by the water molecule’s accessibility in the active site of the FP-2. At last, to understand the geometrical behaviour of the complex structure, Rg analysis was performed. Also, the level of structure compaction was studied using Rg, which shows how the folded and unfolded polypeptide chains are in a complex. The results observed from Rg analysis states that there is a loss of structural activity on the binding of Reticuline, Aclidinium and Artemisinin. On the other hand, Noscapine binding does not make any impact on the structure folding and unfolding. ΔG*_bind_* evaluation can help unambiguously in the identification of the most potent protein inhibitor. The molecular mechanism MMPBSA approach efficiently recapitulates the binding capability of alkaloids to the target protein of FP-2. Noscapine, Reticuline and Aclidinium complexes had stable energy values from the last 20 ns of MD trajectory for instance FP-2-Aclidinium, complex gave the lowest binding free energy averaging at −204.23 +/− 12.37 kJ/mol followed by Reticuline and Noscapine with the binding free energy of −183.37 +/− 10.88 and −104.44 +/− 13.80 respectively, Altogether, the results strongly suggest that Noscapine could be a promising candidate for designing novel FP-2 inhibitors in the future. Interaction analysis also revealed that Noscapine and Reticuline bind to similar amino acid residues (TRP206 and ALA157) of FP-2 protein as that of artemisinin. Most importantly, we have also compared our results with a well-known FP-2 inhibitor that is E64 which has potential FP-2 blocker. It is reported that E64 like antimalarial agents should be capable of H-bond donor and acceptor and also can interact with polar amino acids such as SER41, SER149, ASN138, ASN173, and ASN77 and GLN171 and with charged amino acids such as ASP170, HIS174, and ASP234 of FP-2. The ability of the potent candidate to favourably interact with ASP, ASN, and SER is an important characteristics of a potential candidate [40,51]. This corelates well to current study. Noscapine’s unique pharmacology; impact on cellular signalling pathways, the mitotic spindle, and centrosome clustering, it was suggested to use as an antimalarial drug [52]. However, noscapine has strong anticancer activity (NPACT database), which can be repurposed for antimalarial activity [52] as it has shown a good binding affinity towards FP-2.

## 5. Conclusions

Falcipain-2 is a promising antimalarial therapeutic target because of its essential involvement in the pathogenesis and survival of plasmodia parasites in host erythrocytes. Targeting FP2 with alkaloids is an attractive strategy to combat malaria. The present computational investigation reports that scaffolds of Noscapine can be used in the designing and development of new analogues to target FP-2 protein. Noscapine is a known anticancer drug which can be repurposed as an antimalarial drug target. This is the first inhibitory action of Noscapine towards FP-2 and warrants further in-vivo and in-vitro exploration in the future.

## Figures and Tables

**Figure 1 biotech-11-00054-f001:**
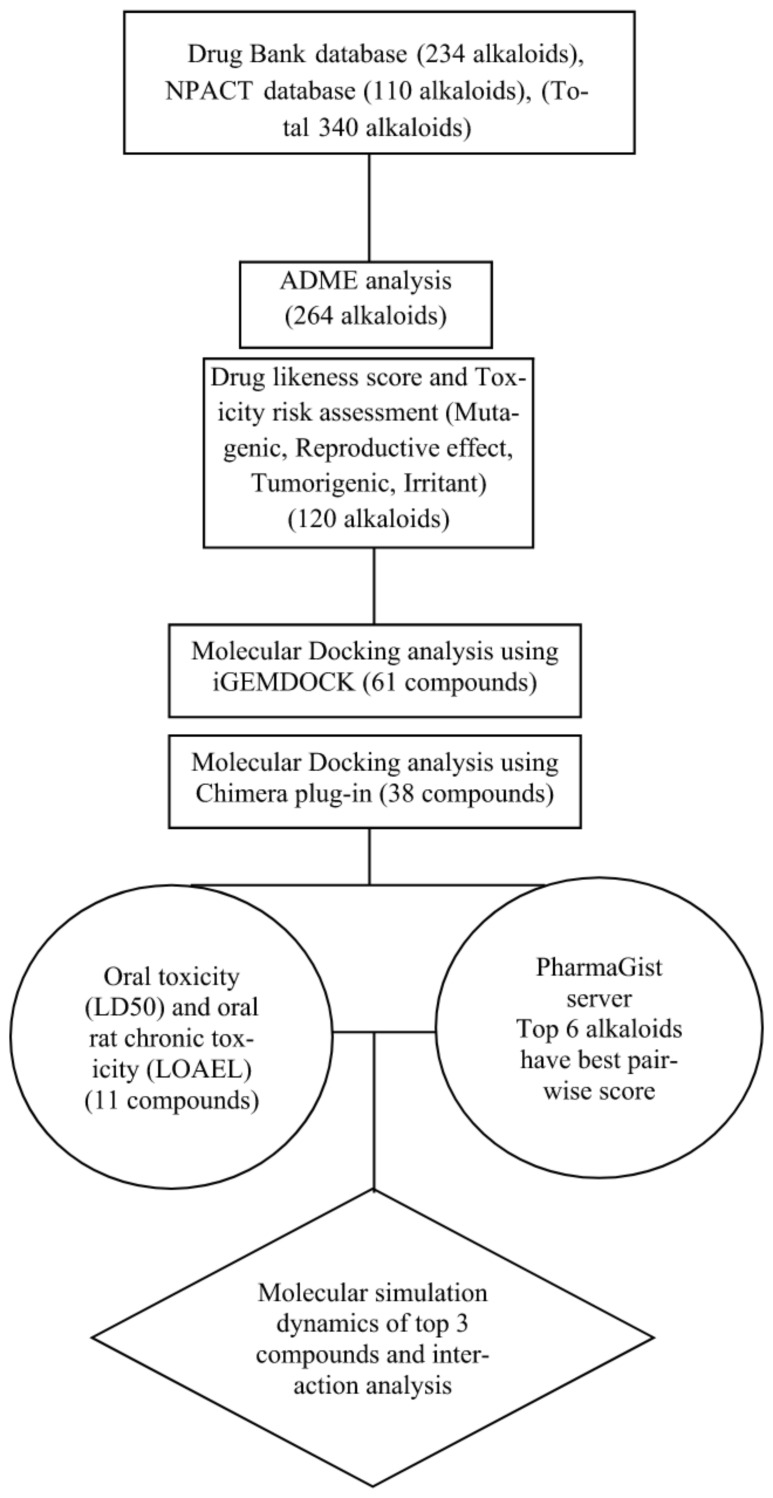
Schematic representation of the work flow adopted in the study.

**Figure 2 biotech-11-00054-f002:**
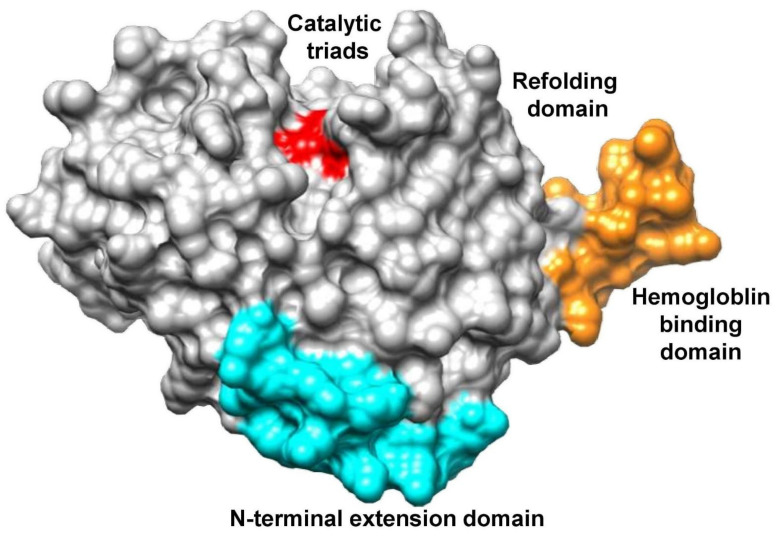
Structure classification of falcipain-2 protein.

**Figure 3 biotech-11-00054-f003:**
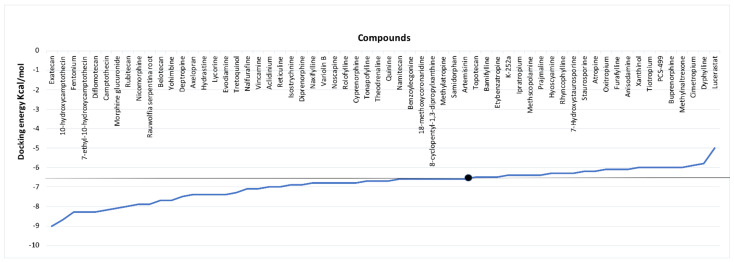
Graphical representation of docking energy scores of screened 62 alkaloids using Auto Dock Vina (Note: Compounds above the horizontal line in the graph indicates the alkaloids with higher binding energies than Artemisinin).

**Figure 4 biotech-11-00054-f004:**
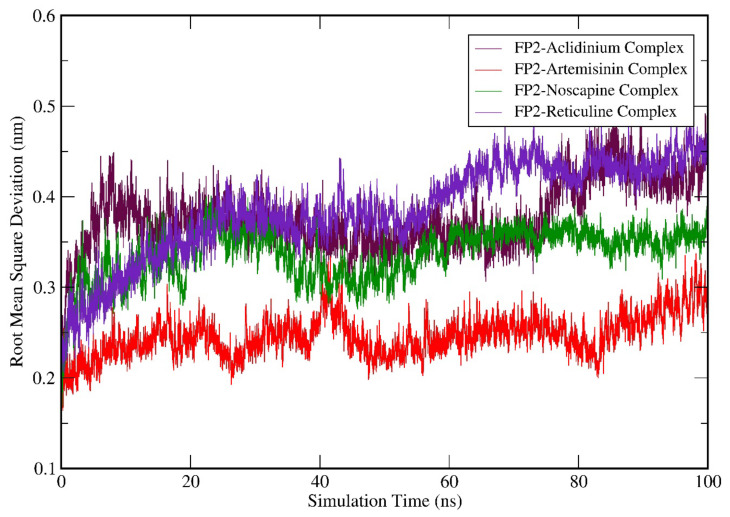
Molecular dynamics simulation (MDS) result analysis for FP-2-Artemisinin, FP-2 Noscapine, FP-2 Reticuline, and FP-2 Aclidinium complexes representing RMSD (Root mean square deviation).

**Figure 5 biotech-11-00054-f005:**
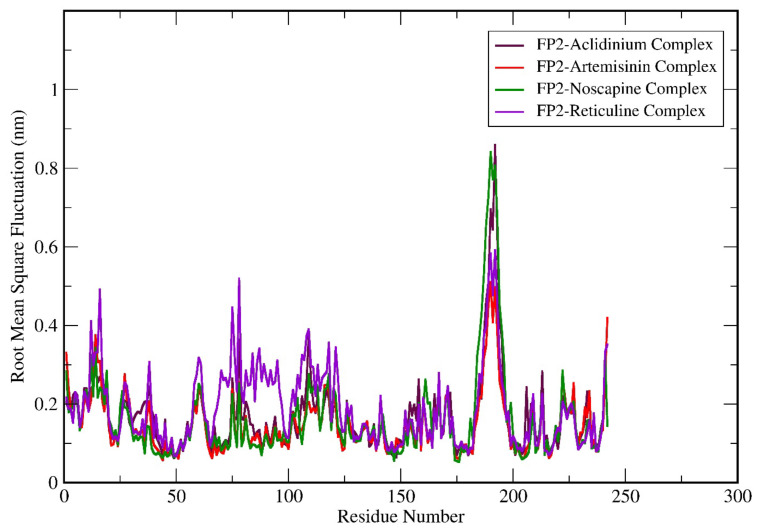
Molecular dynamics simulation (MDS) result analysis for FP-2-Artemisinin, FP-2 Noscapine, FP-2 Reticuline, and FP-2 Aclidinium complexes representing RMSF (Root mean square fluctuation).

**Figure 6 biotech-11-00054-f006:**
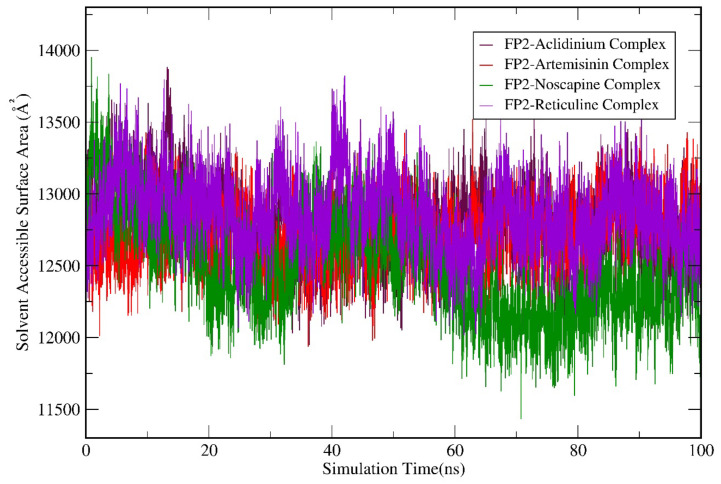
Molecular dynamics simulation (MDS) result analysis for FP-2-Artemisinin, FP-2 Noscapine, FP-2 Reticuline, and FP-2 Aclidinium complexes representing SASA (Solvent-accessible surface area).

**Figure 7 biotech-11-00054-f007:**
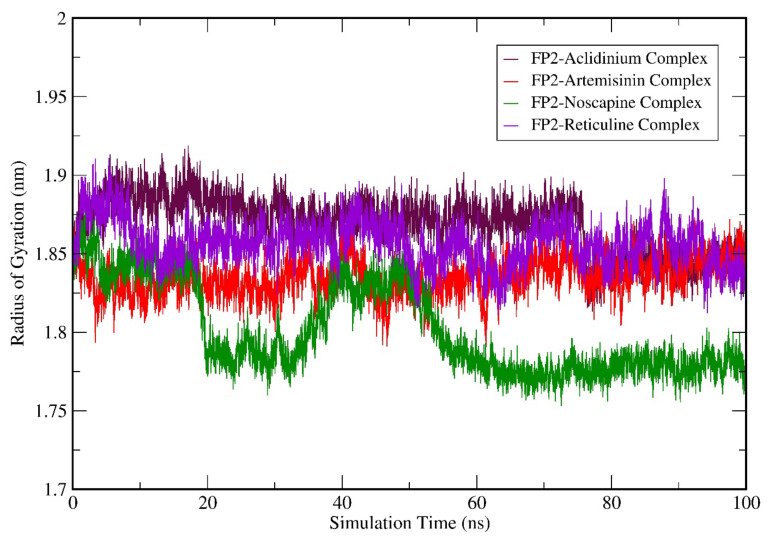
Molecular dynamics simulation (MDS) result analysis for FP-2-Artemisinin, FP-2 Noscapine, FP-2 Reticuline, and FP-2 Aclidinium complexes representing Rg (Radius of gyration).

**Figure 8 biotech-11-00054-f008:**
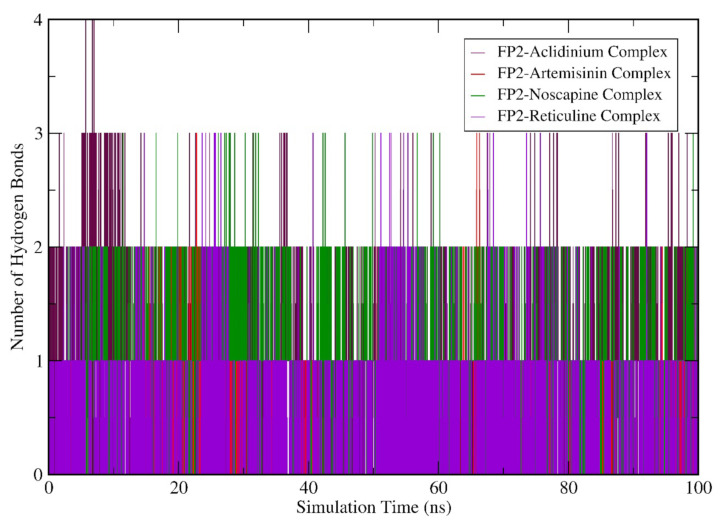
Molecular dynamics simulation (MDS) result analysis for FP-2-Artemisinin, FP-2 Noscapine, FP-2 Reticuline, and FP-2 Aclidinium complexes representing H-Bond (Hydrogen bond).

**Figure 9 biotech-11-00054-f009:**
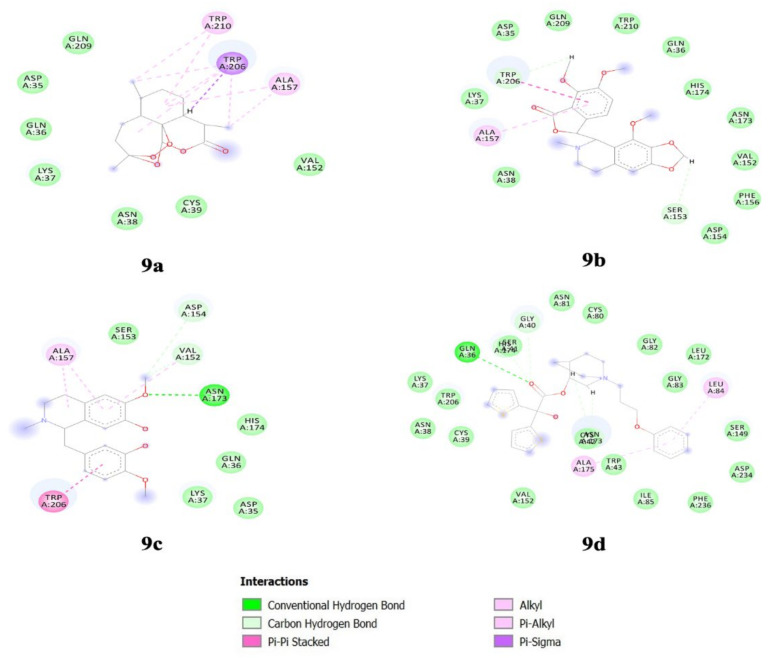
Interacting residues between falcipain-2 and alkaloids (**9a**) Artemisinin (**9b**) Noscapine (**9c**) Reticuline (**9d**) Aclidinium.

## Data Availability

The raw-datasets generated during and/or analysed during the current study are available from the corresponding author on reasonable request.

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
