# Peer review of "Identification of Potential Antimalarial Drug Candidates Targeting Falcipain-2 Protein of Malaria Parasite—A Computational Strategy"

_biotech, 2022, doi:10.3390/biotech11040054_

Round 1

Reviewer 1 Report

The work presented to me for review presents an extremely important issue of the search for an effective antimalarial drug on a global scale. The authors started from the absolutely correct assumption of searching for such substances in the group of alkaloids, which also includes quinine, which is the first antimalarial drug. They conducted an fully comprehensive study using a computational strategy, in which they used, in the first stage, a database of 340 natural alkaloid substances. From them, through several-stage selection, they selected 4 alkaloids with the highest potential for beneficial interaction with the FP-2 protein, which is responsible for the course of most malarial infections.

The regularity and logic of the conducted research is admirable. It leads to an ever-increasing narrowing of the group of selected alkaloids, to finally indicate those 4 worthy of further detailed research using methods of combining them with the appropriate receptor protein. Various, widely available predictive programs are used, but their methodology and sequence of use give an exceptionally beneficial effect. The final molecular dynamics studies of complexes of selected alkaloids with FP-2 are also performed in an excellent way. The description of the research is exhaustive and the results obtained have been skilfully divided between the text of the work itself and supplementary materials and placed in the correct way in the right place.

In the opinion of the reviewer, the only debatable element of the work is the rightness of using ADME calculations with use of Lipinski's rule of 5. There are many commonly used drugs known that do not fully meet all the conditions of this rule. Therefore, such an approach may result in the unintentional elimination of a valuable substance already at the first stage of research. Nevertheless, in this paper, I propose to leave the presented analytical approach unchanged, but only because the authors in their final conclusion indicate the need for further extended research on this issue. In the future, however, modified forms of the famous Lipinski's rule should be used, due to its numerous limitations and imperfections.

Despite this remark, I think that the reviewed paper is respective to be accepted for publication in your journal without any changes or corrections.

Author Response

Response: Thank you for giving us the opportunity to submit a revised draft of the manuscript. We appreciate the time and effort that the reviewer has dedicated to providing feedback on our manuscript and are grateful for the insightful comments on and valuable suggestions to our paper.

Reviewer 2 Report

Antimalarial therapeutic drug discovery efforts including potential effective vaccine development are very much needed. The authors systematically screened about 340 drug candidates using different computational programs and tools. Out of 340 drug candidates, the authors eventually selected the top 3 and performed the MD simulations that show the best binding efficiency. Finally, the noscapine appeared to be a promising candidate to inhibit the FP-2. Overall, the manuscript is written well and is of interest to malaria researchers. I have a few points that may help to improve the manuscript.

1.    The authors selected the top 3 drug molecules based on 3D pharmacophore analysis using PharmaGist based on best pairwise alignment scores. The top 6 scores are close to each other and can be selected for further studies instead of the top 3 unless there are other factors preventing them.

2.    The authors screened a total of 120 alkaloids against FP-2 protein and found 62  have shown greater binding energy scores when compared to Artemisinin. The authors need to describe the scoring criteria and acceptable score range in the methods section to better understand figure 3 without going to ref 27.

3.    Figure 9 interaction analysis looks unclear, it needs more description. What are the distances between the compounds and amino acid residues? what do different colors indicate?

Minor point:

1.    Figure 3 caption- 62 alkaloids instead 42.

2.    Conclusions could be elaborated.

3.    The author may want to cite similar studies- https://doi.org/10.3389/fphar.2022.850176, DOI: 10.2147/DDDT.S265602

Author Response

Response to Reviewer 2:

Antimalarial therapeutic drug discovery efforts including potential effective vaccine development are very much needed. The authors systematically screened about 340 drug candidates using different computational programs and tools. Out of 340 drug candidates, the authors eventually selected the top 3 and performed the MD simulations that show the best binding efficiency. Finally, the noscapine appeared to be a promising candidate to inhibit the FP-2. Overall, the manuscript is written well and is of interest to malaria researchers. I have a few points that may help to improve the manuscript.

Response 1: First of all, let me say thank you for your insightful comments and suggestions, which have helped us greatly improve the quality of our manuscript. We appreciate the reviewer's time and effort in providing comments on our manuscript.

Comment 1: The authors selected the top 3 drug molecules based on 3D pharmacophore analysis using PharmaGist based on best pairwise alignment scores. The top 6 scores are close to each other and can be selected for further studies instead of the top 3 unless there are other factors preventing them.

Response 1: Yes, we agree with the reviewer that the top 6 compounds have the closest score. However, this study has an approach to narrowing down the compounds to get the best possible one. The top 3 compounds have the highest score of the rest of the compounds. Furthermore, the top 3 compounds are anti-cancerous drugs. However, they have antimalarial activity as reported by other researchers (refer to Table). But rest 3 compounds (Benzoylecgonine, Vincamine, and Tretoquinol) are used for different purposes as mentioned in the table, and no antimalarial activity was reported previously. Therefore, we have taken the top 3 compounds for further analysis.

S. No.

Drugs Name

Biological Activity

1.      

Noscapine

Noscapine is widely known as anticancer drug from NPACT database, and can be repurposed for antimalarial activity. Nourbakshsh et. al. has also reported its antimalarial property (https://doi.org/10.1002/biof.1781).

2.      

Reticuline

Reticuline is also anticancer drug from NPACT database and can repurposed for antimalarial activity. However, Uzor et. al. has mentioned about its antimalarial activity (https://doi.org/10.1155/2020/8749083)

3.      

Aclidinium

Aclidinium is also known as anticancer drug. However, Verma et. al. reported its antimalarial activity (DOI: 10.1002/jcb.29954).

4.      

Benzoylecgonine

Benzoylecgonine is the compound tested for in most substantive cocaine drug urinalyses.

5.      

Vincamine

 Vincamine is used for the treatment of primary degenerative and vascular dementia.

6.      

Tretoquinol

Tretoquinol is an adrenergic beta-agonist used as a bronchodilator agent in asthma therapy.

Comment 2: The authors screened a total of 120 alkaloids against FP-2 protein and found 62 have shown greater binding energy scores when compared to Artemisinin. The authors need to describe the scoring criteria and acceptable score range in the methods section to better understand figure 3 without going to ref 27.

Response 2: We have set threshold docking scores of the reference molecule (Artemisinin) for each docking program, iGEMDOCK, and Autodock Vina, i.e., -91.37 and -6.6 kcal/mol, to screen out the alkaloids. The same has been incorporated into the manuscript. [Refer to a line number in methods section – 144-145; result section- 216 to 218]

Comment 3: Figure 9 interaction analysis looks unclear, it needs more description. What are the distances between the compounds and amino acid residues? what do different colours indicate?

Response 3: We appreciate the reviewer’s careful review. We have made a descriptive table for the interaction study (refer to supplementary table S7), which shows the distances between the alkaloids and the interacting amino acid residues. For the readers' better understanding, we have also included the color scheme from Figure 9 (revised).

Figure 9 (revised)

Table S7: Detail of the top compounds with interacting residues, types of bond and bond distance

Compound

Interacting residues

Types of bond with distance

Artemisinin (Reference)

ALA 157

3.78, 4.99 (pi-alkyl)

TRP 206

4.22 (pi-alkyl), 4.13 (pi-alkyl), 4.88 (pi-alkyl and alkyl), 5.09 (pi-alkyl), 2.39 (pi-sigma)

TRP 210

4.67, 5.41 (pi-alkyl)

Noscapine

TRP 206

2.99 (carbon hydrogen), 4.02 (pi-pi stacked), 4.13 (pi-pi stacked)

ALA 157

5.15 (pi-alkyl)

SER 153

2.67 (carbon-hydrogen)

Reticuline

ALA 157

3.93, 4.96 (pi-alkyl),

TRP 206

3.64 (pi-pi stacked)

ASP 154

3.74 (carbon-hydrogen)

VAL 152

5.46 (alkyl), 3.37 (carbon hydrogen)

ASN 173

2.45 (hydrogen bond)

Aclidinium

GLN 36

2.32 (hydrogen bond)

ASN 173

2.38, 2.54 (carbon hydrogen)

HIS 174

2.53 (carbon hydrogen)

GLY 40

2.27 (carbon hydrogen)

ALA 175

4.78 (pi-alkyl)

LEU 84

4.76 (pi-alkyl)

Minor point:

  1. Figure 3 caption- 62 alkaloids instead 42.

Response 1: Thanks for pointing out the typo error. We have made the necessary changes in the revised Figure 3 caption.

  1. Conclusions could be elaborated.

Response 2: As per suggestions, we have elaborated the conclusion section in our revised manuscript (refer to line numbers: 432-444). “Falcipain-2 is a promising antimalarial therapeutic target because of its essential involvement in the pathogenesis and survival of plasmodia parasites in host erythrocytes. Targeting FP2 with alkaloids is an attractive strategy to combat malaria. The present computational investigation reports that scaffolds of noscapine can be used in the designing and development of new analogs to target FP-2 protein. Noscapine is a known anticancer drug that can be repurposed as an antimalarial drug target. This is the first inhibitory action of Noscapine towards FP-2 and warrants further in-vivo and in-vitro exploration in the future”. 

  1. The author may want to cite similar studies- https://doi.org/10.3389/fphar.2022.850176, DOI: 10.2147/DDDT.S265602

Response 3: Thank you for the careful review. We have made the necessary addition to the manuscript (Reference number 43)
